# Facilitating Hotspot Alignment in Tip-Enhanced Raman Spectroscopy via the Silver Photoluminescence of the Probe

**DOI:** 10.3390/s20226687

**Published:** 2020-11-23

**Authors:** Yuan Fan, Dan Jin, Xiuju Wu, Hui Fang, Xiaocong Yuan

**Affiliations:** Nanophotonics Research Center, Shenzhen Key Laboratory of Micro-Scale Optical Information Technology, Institute of Microscale Optoelectronics, Shenzhen University, Shenzhen 518060, China; 2176285320@email.szu.edu.cn (Y.F.); 1810285075@email.szu.edu.cn (D.J.); 2161190203@email.szu.edu.cn (X.W.); xcyuan@szu.edu.cn (X.Y.)

**Keywords:** tip-enhanced Raman spectroscopy, silver photoluminescence, radially polarized laser beam, hotspot alignment, carbon nanotubes, surface-enhanced Raman scattering

## Abstract

A tip-enhanced Raman spectroscopy (TERS) system based on an atomic force microscope (AFM) and radially polarized laser beam was developed. A TERS probe with plasmon resonance wavelength matching the excitation wavelength was prepared with the help of dark-field micrographs. The intrinsic photoluminescence (PL) from the silver (Ag)-coated TERS probe induced by localized surface plasmon resonance contains information about the near-field enhanced electromagnetic field intensity of the probe. Therefore, we used the intensity change of Ag PL to evaluate the stability of the Ag-coated probe during TERS experiments. Tracking the Ag PL of the TERS probe was helpful to detect probe damage and hotspot alignment. Our setup was successfully used for the TERS imaging of single-walled carbon nanotubes, which demonstrated that the Ag PL of the TERS probe is a good criterion to assist in the hotspot alignment procedure required for TERS experiments. This method lowers the risk of contamination and damage of the precious TERS probe, making it worthwhile for wide adoption in TERS experiments.

## 1. Introduction

Tip-enhanced Raman spectroscopy (TERS) is a near-field enhanced Raman spectroscopy imaging technique that can obtain high-resolution topographical image and in-situ nanoscale chemical fingerprint information simultaneously [1]. These advantages make TERS a promising tool for applications in fields where nanoscale surface chemical characterization sensing is required, such as chemical reactions [2,3], materials characterization [4,5], and life sciences [6,7]. Inspired by the principle of surface-enhanced Raman scattering (SERS), when a metalized TERS probe is illuminated by a focused laser beam with appropriate wavelength and polarization [8,9], the electromagnetic field is greatly enhanced because of the excitation of localized surface plasmon resonance (LSPR). Because the enhanced electromagnetic field is confined in the vicinity of the probe apex and decays exponentially along the TERS probe axis [10,11], by scanning a sample across the probe apex and collecting the TERS spectrum at each position, Raman images with high (nanometer-scale) spatial resolution can be obtained.

The quality of plasmon-active metallized probes is an important factor that affects the success of TERS experiments. The preparation of TERS probes with high Raman enhancement ability has been widely discussed [12,13,14]. A common method to fabricate TERS probes is to coat commercially available silicon (Si) AFM tips with a thin layer of silver (Ag) or gold (Au) by physical vapor deposition [15] or electrodeposition [16]. However, because the Ag or Au coating is relatively soft [17] and usually only a few tens of nanometers thick, it is easily worn during TERS experiments in contact mode, which degrades Raman enhancement ability. Although an ultrathin layer of aluminum oxide (Al_2_O_3_) [17,18] or zirconia (ZrO_2_) [19] can be added to extend the lifetime of such probes, precautions still need to be taken to avoid unnecessary abrasion during the complex and time-consuming hotspot alignment process. Frequent inspection can identify and avoid TERS probe wear in TERS experiments; however, there still a lack of good methods that can routinely track the plasmon enhanced electromagnetic field changes of TERS probe during experiment.

A critical step in TERS experiments is to align the TERS probe with the focused laser spot to produce an electromagnetic hotspot at the tip apex, which we hereafter call “hotspot alignment”. Only when the tip-laser coupling is well realized can the TERS probe exert its maximum electromagnetic field enhancement performance. Fast and accurate hotspot alignment can minimize the risk of TERS probe wear. At present, laser beam scanning across tip or tip scanning across laser focal spot are two alternative ways to align the hotspot. Scanning the laser beam has the advantage that no tip movement is needed, thus protecting the probe from damage during the hotspot alignment. However, this method increases the complexity of the system because an additional auxiliary component such as a galvanometer scanner is required [20,21,22]. The custom-built system we report here is based on tip scanning. Three methods of hotspot alignment based on tip scanning have been reported to date. The first is to scan the tip just across the laser focal spot to reconstruct the intensity distribution of the tip scattering signal acquired by an extra photomultiplier tube (PMT) [23,24] or avalanche photodiode (APD) [25,26]. However, the resulting tip scattering signals contain Rayleigh scattering, the photoluminescence (PL) of the Ag coating, and even the stray light from the optical system, which may weaken the hotspot mapping contrast. The second method is to scan the probe across a homogeneous sample layer on a glass coverslip and generate a Raman image of the sample. A uniformly distributed sample is required in this case, which increases the complexity of sample preparation. The third method was recently proposed by Kumar et al. [27], who used the PL intensity of the Ag coating on the TERS probe in the spectral range of 100–250 cm^−1^ to locate the hotspot. Among these three hotspot alignment methods, using the PL of an Ag coating is attractive for our TERS system because it is easy to implement and the PL of Ag can be collected by Raman spectroscopy.

The PL of bulk metals is very weak because of their low probability of corresponding interband radiative transitions [28]. However, the PL of Ag or Au nanostructures is widely observed in SERS spectra as a broad and continuous background because of the strong LSPR enhanced electromagnetic field [29]. This wide range of metal PL background existing in raw SERS spectra receives little attention and is usually removed by precise baseline correction [30] to extract the Raman signals of interest. Nevertheless, the PL of metal nanoparticles contains useful local electromagnetic field intensity information and also exists in TERS. Here we take advantage of the specificity of the internal electromagnetic field contained in the PL intensity of Ag-coated TERS probe and extend its application to TERS experiments. In addition, to accelerate the initial alignment of the apex of the TERS probe relative to the laser focal spot, we optimize the optical path in our TERS configuration.

## 2. Experiment Setup

A schematic diagram of our TERS experimental setup is shown in Figure 1a. The setup consisted of an AFM (NT-DMT, NTEGRA, Moscow, Russia), inverted microscope platform (IX71, Olympus, Tokyo, Japan), laser (532 nm, Ventus, Konstanz, Germany), and Raman spectrometer (iHR550, Horiba, Kyoto, Japan) equipped with a liquid nitrogen-cooled Symphony II (Horiba, Kyoto, Japan) charge-coupled device (CCD) detector. The AFM head was positioned on the inverted microscope platform and a traditional bottom-illumination configuration was used. The laser was expanded first and its polarization state was controlled by passing sequentially through a linear polarizer, half-wave plate (WPH05M-532, Thorlabs, Newton, NJ, USA), and vortex half-wave retarder (WPV10L-532, Thorlabs, Newton, NJ, USA). After reflection by two beam splitters (BSs), the laser beam was directed into the inverted microscope and focused by an immersion oil microscope objective (numerical aperture NA = 1.49) on the sample prepared on a glass coverslip (CG15CH2, Thorlabs, Newton, NJ, USA). The excited Raman signal was collected by the same microscope objective and focused by the tube lens into a confocal pinhole (100 μm) placed in the front focal plane of the tube lens to reject possible stray light. After passing through the pinhole, the Raman scattering light beam was then collimated by lens3, reflected by two mirrors, and finally directed by lens4 into the slit of the Raman spectrometer. To ensure the Raman signal coupling efficiency and easy alignment of the light path, the NA of the beam entering the spectrometer slit was controlled to be slightly lower than that of the spectrometer. The spectrometer slit width was set to 20 μm, which corresponded to a confocal diameter of about 600 nm on the sample plane. To block the Rayleigh scattering light, two long-pass filters, LPF1 (RL-532, Shanghai-optics, Nanjing, China) and LPF2 (LP03-532RE-25, Semrock, Rochester, NY, USA) were added, which were located on different sides of the pinhole. This carefully designed confocal collection optical path minimized the interference from stray light. As a result, the measured Raman spectrum had a very flat baseline. The AFM system only provided white-light illumination from above, which was only suitable for viewing the apex protruding from the end of the TERS probe cantilever. Therefore, a halogen lamp (HL) was included to enable illumination from the bottom. The light from the HL was coupled into an optical fiber with a diameter of 100 μm and then collimated by lens1. After being reflected by two BSs, the light emitted from the optical fiber was imaged on the back focal plane of the microscope objective and uniform bottom illumination was obtained at the focal plane of the microscope objective. As illustrated in Figure 1e, the TERS probe apex and laser focal spot can be seen simultaneously from camera1, which facilitated the preliminary alignment of the tip and laser.

Laser polarization strongly influences TERS performance [31]. Strong longitudinal electric field components can be acquired by a tightly focused radially polarized laser beam, which is beneficial to the LSPR excitation of a TERS probe in our transmission-mode TERS configuration. Two methods were used to precisely calibrate the polarization state of the laser. One was to place a glass coverslip coated with a 50 nm-thick Ag film on the sample stage, rotate the HWP and observe the reflected back focal image from camera2. As shown in Figure 1b, when a complete dark ring appears, the laser is radially polarized [32]. As depicted in Figure 1c, when the dark ring completely disappears, the laser is azimuthally polarized. When the laser was linearly polarized, the back focal image is shown as in Figure 1d. The second method was to add a flip LP and observe the splitting direction of the laser focal spot [33] from camera1, as displayed in the insets of Figure 1b,c. An optical power meter was used to measure the laser power entering the entrance pupil of the microscope objective. To avoid heat damage to the TERS probe, the laser power was kept below 300 μW.

Data analysis: All spectra were processed in MATLAB (version R2018b, MathWorks, Natick, MA, USA). Spectra preprocessing included baseline correction and removal of cosmic ray strikes.

## 3. TERS Probe Preparation

The electromagnetic field enhancement capability and robustness are the two most important considerations when producing TERS probes. The characteristic plasmon resonance wavelength (PRW) of the TERS probe and wavelength of the excitation laser need to match to generate LSPR near the apex of the TERS probe and maximize the electromagnetic field enhancement. Commercially available Si AFM probes (CSG10, NT-MDT, Moscow, Russia) operating in contact mode were used in our experiments. Because of its high refractive index *n* of 4.15 at 532 nm, if an Ag film is directly coated on the Si probe, its PRW will shift to longer wavelength from 532 nm. Therefore, it is recommended to carry out refractive index modification, as reported previously [15,34]. Oxidation of the Si substrate of the probe can lower its *n* but lengthens the probe preparation process. Adding a layer with low *n* on Si substrate of the probe is an alternative way to modify *n*, although it will increase the radius of probe apex, which results in a sacrifice of spatial resolution. Al_2_O_3_ is a tip protection material frequently used in TERS probes that can extend their lifetime up to 40 days because of its high hardness and compactness [17]. Moreover, Al_2_O_3_ is transparent with a low *n* of 1.77 at 532 nm. We found that by precoating only 10 nm-thick Al_2_O_3_
*n*-modification layer on the Si probe substrate can shift the PRW of the TERS probe from the yellow region to the green region. Dark-field microscopy (BX51M, Olympus, Tokyo, Japan) was used to estimate the PRW of the newly prepared TERS probe [35]. A dark-field micrograph of the bare Si probe coated with only a 45 nm-thick Ag film is shown in Figure 2a. This image indicates that the PRW is in the yellow region. After precoating with 10 nm of Al_2_O_3_, the PRW shifted to the green region, as illustrated in Figure 2b. Therefore, *n*-modification with 10 nm-thick Al_2_O_3_ improved the match of the PRW of the probe with the excitation wavelength.

The TERS probe was fabricated as follows. Electron-beam evaporation (Syskey Technology Corporation, Hsinchu, China) was used to prepare the TERS probe under a high vacuum of 2 × 10^−4^ Pa. Four layers were coated sequentially on the Si substrate without breaking the vacuum conditions: 2 nm of chromium (Cr) as an adhesion layer, 10 nm of Al_2_O_3_ as an *n*-modification layer, 45 nm of Ag as a plasmon layer, and then 3 nm of Al_2_O_3_ as a protective layer. The TERS probe without an *n*-modification layer was prepared as described above except that the 10 nm-thick Al_2_O_3_ layer was omitted.

An SEM image of the prepared TERS probe by vacuum-deposition of silver on AFM probe after an *n*-modification layer is shown in Figure 2c. The Ag film coated on the surface of the probe appears rough. A single Ag particle with a diameter of about 50 nm was observed at the apex of the probe. The SEM image of Si AFM probe is presented in Figure 2d.

## 4. Results and Discussion

In TERS imaging, the probe should maintain high plasmon enhancement activity for a long time. However, the plasmon enhancement capability of the TERS probe gradually decrease over time, especially after the hotspot is generated at the apex of the probe. Because of the near-field thermal effect [36] and the force applied to the apex of the probe, the oxidation and wear rate of the Ag film on the probe apex will accelerate after hotspot generation. To minimize the deterioration of the probe’s plasmon enhancement capability, we need to optimize the probe coating parameters, AFM feedback parameters, and laser power. During the process of optimizing the probe coating and TERS system parameters, we found that the PL of the Ag coating of the TERS probe can act as a good aid for tracking the electromagnetic field change at the apex of the TERS probe. Once the probe is worn, oxidized, or undergoes mechanical drift relative to the laser spot, the enhancement of the electromagnetic field near the apex of the TERS probe will weaken. We found that the PL intensity of Ag on the TERS probe decreased simultaneously. Therefore, monitoring the PL intensity of the Ag layer of the TERS probe will enable damage or drift to be detected as early as possible and allow improvements to be made.

To confirm the PL characteristics of the Ag-coated TERS probe, a newly prepared Ag-coated TERS probe was coupled with the focused laser spot to produce a hotspot at the probe apex; the PL of the TERS probe acquired is shown in Figure 3a (Tip in). The background spectrum (Tip out), which is also shown in Figure 3a, was acquired when the TERS probe was retracted. The two spectra in Figure 3a were obtained with only the optical filter LPF2 in the TERS optical system. The transmittance curve of LPF2 is shown in Figure 3b and the start transmittance wavenumber is about 90 cm^−1^. The background spectrum (Tip out) is a relatively flat line. But its intensity usually fluctuated between 700 and 800 counts in our system over time, mainly because of the fluctuation of readout noise and thermal noise of the spectrometer CCD. The PL intensity of the Ag-coated TERS probe is in a decline trend in 100 to 400 cm^−1^ and gradually leveled off after 400 cm^−1^. Therefore, it is easy to mistake this signal as the background caused by CCD noise. However, we believe that the difference between the two spectra (Tip in and Tip out) in Figure 3a should mainly originate from the PL of Ag excited by LSPR at the apex of the TERS probe. The Ag PL in the range of 100–250 cm^−1^ is stronger than that at higher wavenumber, but there is no obvious peak. To observe and utilize the slight change of the PL intensity of the Ag-coated TERS probe more intuitively, and eliminate the influence of background noise fluctuation on the Ag PL intensity, another optical filter (LPF1) was added, which was used to select specific bands from the Ag PL spectrum and create a clear artificial peak at 195 cm^−1^. The transmittance curve of LPF1 is shown in Figure 3b. The transmittance of LPF1 at 195 cm^−1^ is about 80%. In the following experiments, both optical filter LPF1 and LPF2 were present in the experimental setup. The Ag PL peak at 195 cm^−1^ we mention in the subsequent experiments is not the real peak of Ag PL, but the peak filtered by LPF1. The presence of optical filter LPF1 was not always necessary. The PL intensity of Ag in the range of 100 to 250 cm^−1^ is also suitable to characterize the Ag PL, but an accurate background must be measured and subtracted each time. At this time, the traditional baseline correction method is not recommended because it would lose most of the Ag PL information. If the Raman range of tested sample is less than 500 cm^−1^, LPF1 should be removed.

After fabrication of the TERS probe, we needed to evaluate its electromagnetic field enhancement capability and stability. Without testing the Raman samples, we found that the PL intensity of the Ag-coated TERS probe is an effective indicator to check the stability of the enhanced electromagnetic field of the TERS probe upon laser irradiation. According to our experience, the test data we obtained of a good TERS probe is now presented. The apex of a TERS probe was first aligned with the laser spot to produce a hotspot at the apex, then 400 spectra of the probe were recorded continuously by the spectrometer and are displayed as a time series waterfall plot in Figure 3c. The temporal intensity fluctuation of the Si Raman peak at 520 cm^−1^ and Ag PL at 195 cm^−1^ from the TERS probe were extracted and are shown in Figure 3d. Among the 400 spectra in Figure 3c, about 133 spectra showed different degrees of the blinking phenomenon. “Blinking” of the Ag-coated probe is spectral intensity fluctuations caused by molecules [37] or Ag nanoclusters [38,39] diffusing in and out the strong gradient field [40,41] near the probe apex. Ten selected characteristic blinking spectra taken at different times are presented in Figure 3e. Although the mechanism of blinking is still a subject of debate, the continuous blinking indicates that the prepared TERS probe has a stable and high electromagnetic enhancement capability because theoretical calculations have predicted that at least ~10^8^ enhancement is necessary to observe blinking [42]. The temporal fluctuation of PL intensity (Figure 3d) suggests that no obvious decrease of the electromagnetic field near the apex occurred during the strong blinking, which also indicates the high electromagnetic enhancement capability and stability of this TERS probe. Moreover, we found that blinking usually brought about a short-term increase in the PL intensity of Ag, as shown in Figure 3d, which indicates the existence of new intense sub-nanoscale hotspots formed at the apex of the Ag-coated TERS probe by the diffusion of Ag nanoclusters during blinking. This can be explained by what Lindquist et al. [39] have reported that thermal and photoinduced processes may play a role since local heating can accelerate structural changes, modify the resonance conditions of hotspot or boost the formation of sub nanometer cavities. The positions of these blinking peaks were random, so we averaged all 400 spectra, as shown in Figure 3e; no obvious Raman peaks were observed in the averaged spectrum. Therefore, we do not think that the probe is contaminated by organic molecules. The blinking phenomenon of the TERS probe can be alleviated when the laser power is reduced [39]. We suggest that the laser power should be reduced until the blinking phenomenon disappears before implementing TERS imaging. If the PL intensity of the TERS probe decreases obviously over time, the protection of the TERS probe needs to be improved or the laser power needs to be reduced. The temporal fluctuation of Si Raman intensity can be used to evaluate the stability of AFM feedback and improve vibration isolation measures.

A smooth Ag film coated on a glass coverslip also emit PL under 532 nm laser excitation, but its intensity was much lower than the PL of the Ag coating on the AFM probe because of the lack of near-field electromagnetic enhancement. We put a glass coverslip coated with a 50 nm-thick Ag film on the microscope sample stage and measured its PL spectra at different laser powers. The PL intensity of the Ag film on the glass coverslip exhibited a linear relationship with excitation laser power from 6 to 4000 μW, as shown in Figure 3f. The inset of Figure 3f depicts all PL spectra of this Ag film (Ag PL peak filtered by LPF1) obtained at different laser powers. Another purpose of this test is to determine the maximum laser power at which the Ag film coated on the glass coverslip will damage. When the laser power exceeds 5000 μW, the PL intensity of the Ag film coated on glass coverslip will decrease which indicates that the Ag film was damaged. Because of the near-field electromagnetic enhancement, the Ag-coated TERS probe is more susceptible to damage by high laser power than the smooth Ag film. Based on our experiment experience, the maximum laser power (532 nm) should not exceed 50 and 200 μW respectively, corresponding to Ag-coated probes without and with the thin Al_2_O_3_ protective layer. We observed that the damage to the TERS probe caused by excessive laser power is permanent. And when the Ag-coated TERS probe is coupled with a laser (laser power low enough) to produce a hotspot, the greater the electromagnetic field obtained at the apex of the probe by the excitation of LSPR, the greater the PL intensity of Ag from the TERS probe. Therefore, the PL intensity of Ag should be a suitable criterion to investigate tip-laser coupling to form a hotspot at the probe apex. Next, we conducted two experiments to determine whether it is feasible to use the PL of Ag to realize the tip-laser coupling that signifies hotspot alignment at the TERS probe apex.

The first experiment performed to check the suitability of the PL of Ag as an indicator of hotspot alignment involved tip scanning across radially and azimuthally polarized laser focal spots. Under HL illumination, the apex of the TERS probe was found first and moved to the center of the focused laser spot. Then, by raster scan the probe apex across the laser focal spot and reconstructed the intensity distribution and the hotspot patterns from the tip spectra collected by Raman spectrometer. The intensity distribution maps of the laser focal spot under radial polarization and azimuthal polarization reconstructed using the intensity of the 520 cm^−1^ peak of Si are presented in Figure 4a,c, respectively, and the corresponding hotspot patterns reconstructed using the intensity of the 195 cm^−1^ peak (PL peak of Ag filtered by LPF1) are shown in Figure 4b,d, respectively. The intensity distributions in Figure 4a,c are slightly larger than it actually is which can be attributed to the convolution of the exposed pyramidal Si tip with the laser focal spot. Figure 4a,b reveal that the hotspot mapping center does not coincide with the center of the light field under radial polarization. This is because the laser polarization direction relative to the probe is the main factor generating the hotspot rather than the laser intensity. The actual TERS probe has a certain asymmetry and is not oriented completely perpendicular to the glass coverslip. Yeo and co-workers [43] proposed that varying the tilt angle of a metallized tip with respect to the sample surface may affect the near-field Raman enhancement. Under azimuthal polarization, the hotspot mapping presented a ring shape. The maximum Ag PL intensity in the hotspot mapping under radial polarization was about four times stronger than that under azimuthal polarization when the apparent blinking point in Figure 4d was ignored. Therefore, we judged that the PL of the Ag film is a suitable criterion to identify hotspot alignment. The metal PL is also a suitable criterion to determine hotspot alignment for Au-coated TERS probes with different excitation wavelengths and different TERS structures.

Then, we used the developed tip–laser coupling method employing the PL of Ag in actual TERS imaging of single-walled carbon nanotubes (SWCNTs). The SWCNTs suspended in water (TNSR, TIME&NANO, Chengdu, China) were diluted using ultrapure water to the concentration lower than 1 × 10^−4^ wt% (percentage by mass), dropped and spread over a clean glass coverslip, and then dried in air. The glass coverslip was washed thoroughly with distilled water and irradiated with UV/ozone to remove organic matter before use.

The TERS imaging process was conducted as follows. The laser was tuned to radial polarization and the laser power was adjusted to about 100 μW. First, we found the apex of the TERS probe (probe-1) through camera1, as shown in Figure 1e, and then the apex was moved to the center of the focused laser spot. This process can be completed in less than a minute. According to previous experimental experience, the apex position of the probe was now less than 0.5 μm away from the position where the maximum PL intensity of the Ag-coated TERS probe was achieved. Then, the probe scanning mode was selected, the scanning range was set to 0.5 μm × 0.5 μm, the step size was set to 25 nm, and the scanning rate was set to 0.5 s/point. The probe apex was scanned across the focused laser spot and the PL intensity distribution at 195 cm^−1^ was reconstructed. Next, the apex of the probe was moved to the position with the highest PL intensity to obtain the optimal tip–laser coupling to produce a hotspot. This process took about 7 min. To avoid damage caused by the near-field hotspot and reduce the far-field Raman signal of the SWCNTs, after the hotspot was aligned, the laser power was lowered to about 8 μW. The apex position of the probe was kept still and the AFM was switched to the sample scanning mode. The SWCNTs were then scanned across the apex of the TERS probe with a hotspot and the TERS spectrum at each position was recorded. The obtained TERS image is shown in Figure 5a and the corresponding AFM morphology of SWCNTs acquired simultaneously is depicted in Figure 5b. The TERS image of the SWCNTs was reconstructed using the intensity of the G peak (1591 cm^−1^). Figure 5c displays the intensity profile obtained along the purple line of the TERS image in Figure 5a. A Gaussian fit indicated that the spatial resolution was about 28 nm, which is close to the known radius of the probe apex. The actual resolution is higher if the diameter (9 nm) and tilt of the SWCNTs are considered. The resolution of the TERS image can be improved by using a sharper TERS probe and improved system vibration isolation measures. Figure 5d shows the TERS spectra of SWCNTs at two positions indicated in Figure 5b, P1 and P2 marked in the AFM image stand for positions. The successful acquisition of a TERS image of SWCNTs with the resolution close to the radius of used TERS probe confirms that the PL of Ag is a suitable criterion to assist in hotspot alignment.

Figure 5e presents a TERS spectrum (Tip-in) of SWCNT using another TERS probe (probe-2) which was the in the same batch used in the TERS imaging described above. And the far-field spectrum (Tip-out) was acquired when the tip was retracted. The “contrast” calculated is about 20 using the equation [44] defined as:(1)$$Contrast=\frac{I_{Tip-in}}{I_{Tip-out}}-1.$$

The TERS spectrum of SWCNT was acquired as follows. The hotspot alignment process was the same as described above. After the hotspot alignment, the laser power was set to about 8 μW and the sample stage was controlled to move gradually step by step (10 nm) until the tip encountered a SWCNT. The TERS spectrum of SWCNT was then recorded (0.5 s), and the far-field spectrum (0.5 s) was recorded when tip retracted. We noticed that in the TERS spectra of SWCNTs presented in Figure 5d, the intensity of silver PL at 195 cm^−1^ (peak filtered by LPF1) is lower than that measured using another TERS probe (probe-2) in Figure 5e. There may be two reasons. First, the plasmon electromagnetic enhancement ability of probe-1 is lower than that of probe-2. Secondly, during the TERS imaging process, the scanning of the sample stage may aggravate the jitter of probe-1, thereby causing a decrease in its plasmon electromagnetic enhancement ability.

Our customized TERS system also greatly lowers the risk of probe wear and contamination because only one probe scan is needed and the scan area is only 0.5 μm × 0.5 μm. Using the PL of Ag as a criterion to realize tip–laser coupling to produce hotspot is convenient and quick; the hotspot alignment process took less than 8 min.

## 5. Conclusions

An AFM-based TERS system integrated with a radially polarized laser beam was established. The optical path of the TERS system was specially designed. The laser focal spot and TERS probe apex could be seen simultaneously in our setup, which improved the initial alignment accuracy of the probe with the laser, thereby decreasing the wear risk of the TERS probe caused by probe scanning over a wide range. We proposed an easy method to evaluate the stability of the plasmon enhancement capability of the probe by tracking the Ag PL intensity change of the TERS probe, which can also be used to quickly detect the damage of the TERS probe. The successful TERS imaging of SWCNTs confirmed that the PL intensity of Ag is a good criterion to realize quick and accurate tip–laser coupling to produce a hotspot at the apex of a TERS probe.

## Figures and Tables

**Figure 1 sensors-20-06687-f001:**
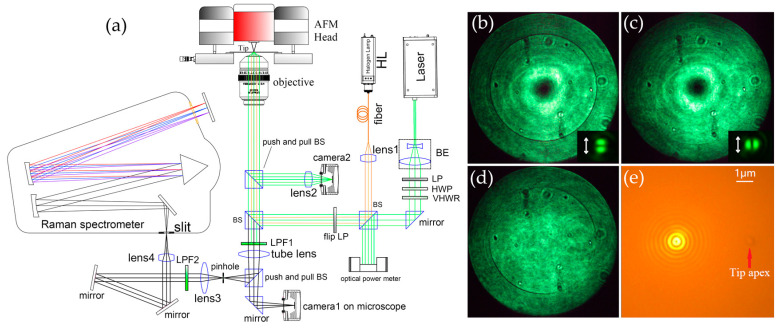
(**a**) Schematic of our atomic force microscope (AFM)-based tip-enhanced Raman spectroscopy (TERS) setup. Intensity distribution at the back focal plane of the microscope objective (camera2) after reflection when the polarized laser (**b**) radially polarized, (**c**) azimuthally polarized, and (**d**) linear polarized is focused on a glass coverslip coated with a 50 nm-thick Ag film. The dark ring corresponds to the surface plasmon excitation. The insets in (**b**,**c**) (camera1) show the splitting direction of the laser focal spot of the corresponding polarization after passing through an additional flip linear polarizer (LP). The direction of the transmission axis of the LP is indicated by a white arrow. (**e**) Image captured by camera1 showing the laser focal spot and TERS probe apex simultaneously. Abbreviations: BS: Beam splitter, BE: Beam expander, LP: Linear polarizer, HWP: Half-wave plate, VHWR: Vortex half-wave retarder, LPF: Long-pass filter, HL: Halogen lamp.

**Figure 2 sensors-20-06687-f002:**
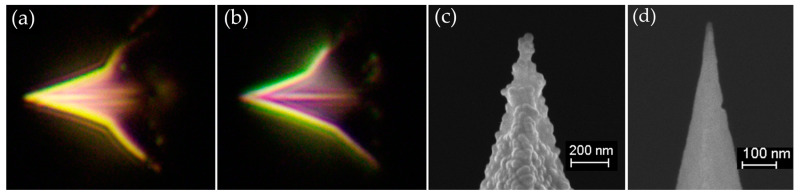
Dark-field images of TERS probes (**a**) without refractive index modification and (**b**) with refractive index modification by precoating with a 10 nm-thick layer of Al_2_O_3_. (**c**) SEM image of the refractive index-modified Ag-coated TERS probe. The Ag particle diameter at the tip apex is about 50 nm. (**d**) SEM image of a Si AFM probe before Ag coating.

**Figure 3 sensors-20-06687-f003:**
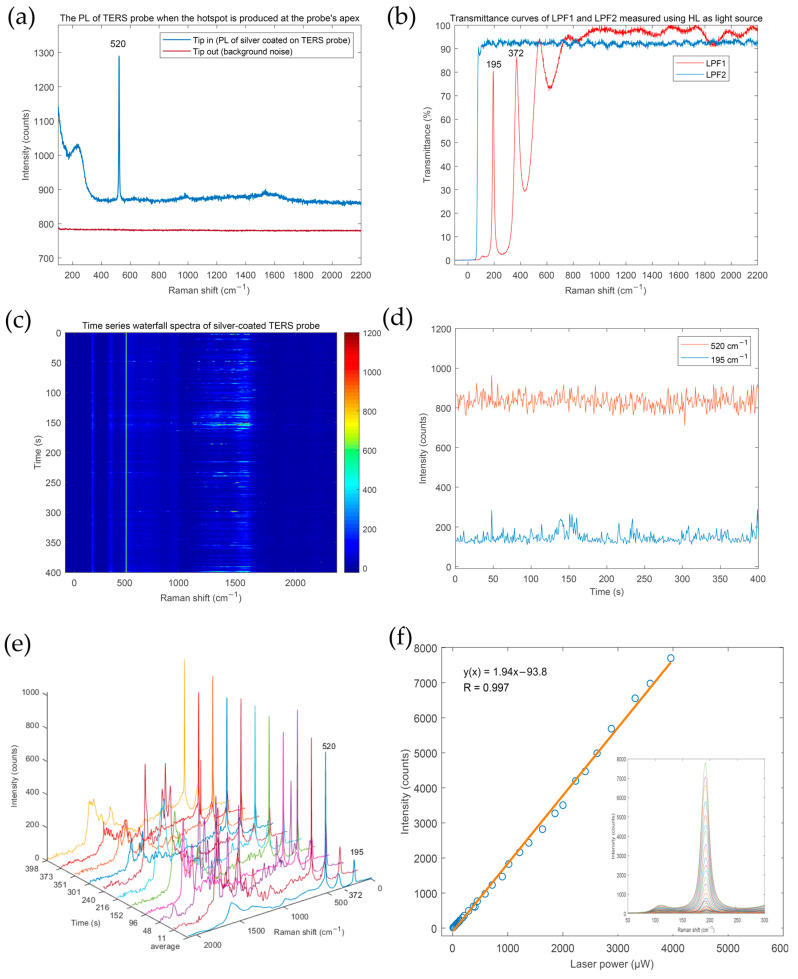
(**a**) Photoluminescence (PL) spectrum of an Ag-coated TERS probe when a hotspot is produced at the probe apex. (**b**) Transmittance curves of LPF1 and LPF2 measured using a halogen lamp as a light source. (**c**) Time series waterfall plot spectra of the Ag-coated TERS probe taken continuously when a hotspot is produced at the probe apex (laser power: 100 μW, integration time: 1 s per spectrum). (**d**) Temporal intensity fluctuation of the Si Raman peak (520 cm^−1^) and PL (195 cm^−1^) of the Ag-coated TERS probe extracted from (**c**). (**e**) Ten selected blinking spectra of the Ag-coated TERS probe extracted from (**c**) taken at different times and the average spectrum. (**f**) Relationship between the PL intensity at 195 cm^−1^ of a 50 nm-thick Ag film coated on a glass coverslip and laser power (integration time: 10 s, baseline corrected).

**Figure 4 sensors-20-06687-f004:**
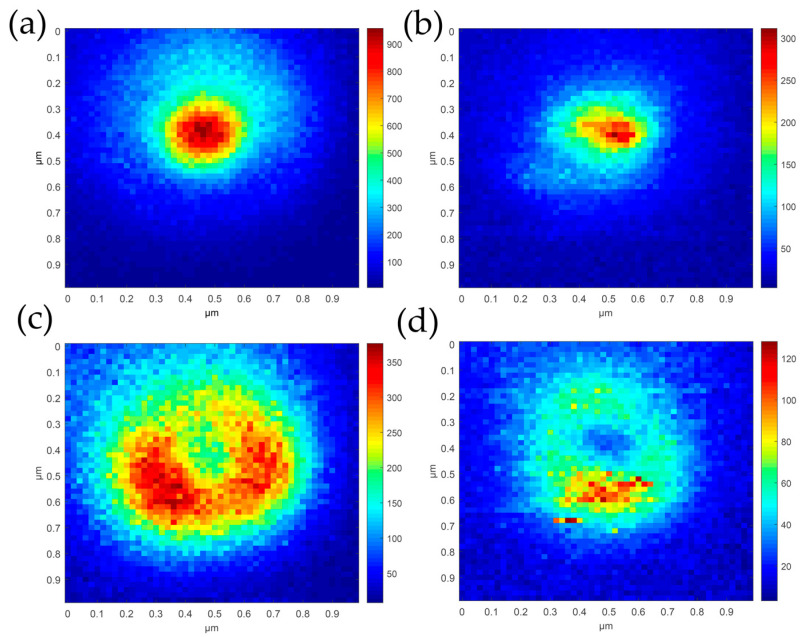
Intensity distributions of the tightly focused laser spot with (**a**) radial polarization and (**c**) azimuthal polarization reconstructed using the intensity of the Si Raman peak at 520 cm^−1^ by raster scans of the TERS probe across the laser focal spot. Corresponding hotspot mapping patterns under (**b**) radial polarization and (**d**) azimuthal polarization reconstructed using the Ag PL intensity. Laser power: 180 μW, 50 × 50 pixels, exposure time of each pixel: 0.5 s. The results were obtained from the same TERS probe. Azimuthal polarization was scanned before radial polarization. Each scan took 50 min.

**Figure 5 sensors-20-06687-f005:**
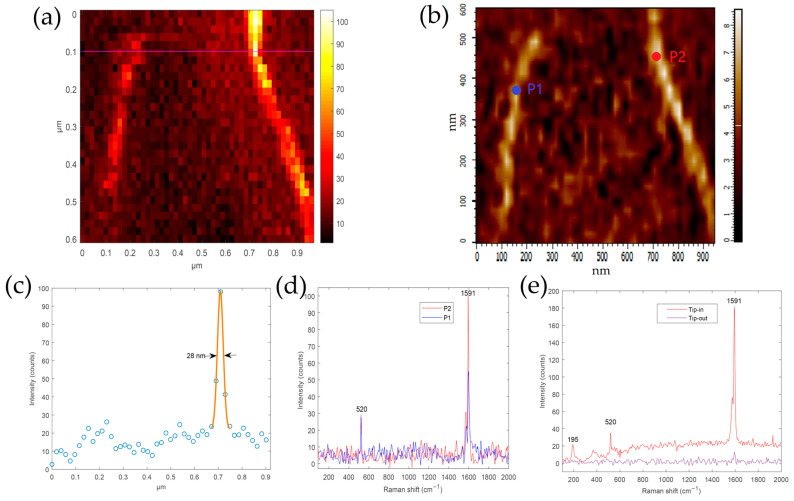
(**a**) TERS image and (**b**) AFM morphology of single-walled carbon nanotubes (SWCNTs) measured simultaneously. The TERS image of the SWCNTs was reconstructed from the G peak (1591 cm^−1^). Exposure time of each pixel: 0.5 s, 50 × 30 pixels, laser power: 8 μW. (**c**) Height profile taken along the purple line in (**a**), which indicates that the spatial resolution of the TERS experiment is about 28 nm. (**d**) TERS spectra of SWCNTs at two positions indicated in (**b**). (**e**) TERS spectrum and far-field spectrum of SWCNT using another TERS probe.

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
