# Peer review of "Facilitating Hotspot Alignment in Tip-Enhanced Raman Spectroscopy via the Silver Photoluminescence of the Probe"

_sensors, 2020, doi:10.3390/s20226687_

Round 1

Reviewer 1 Report

This article developed an optical system to support easy alignment for tip-enhanced Raman spectroscopy (TERS). The results can be summarized as below:

  • An optical system converted the linear polarized laser to the radial polarized laser light, which was confirmed by the SPR resonance of a 50 nm Ag-coated glass plate.
  • An Ag-coated probe was prepared by electron beam deposition of Ag on commercial Si needle which is precoated by Cr and Al2O3 for adjustment of refractive index.
  • The laser irradiation system is hybridized with a halogen lamp illumination to facilitate matching the laser spot and the Ag-coated probe position.
  • The probe position was coarsely adjusted under visible light microscopic image, and optimized by intensity of photoluminescence of Ag, which indicated the hot spot sharper than the Si Raman signal.
  • After optimization of probe position, as a demonstration, a TERS/AFM image of SWCNT was measured.

In my opinion, this system is rationally designed. The alignment of laser and probe are important, and easy alignment is widely required. The process is reasonable and easily understood. However, additional information is needed to justify this technique. In this manuscript, the “successful examples” are reported, but the “limit” is not clear. Some comparison can improve the quality of this manuscript more. I listed the issues to be solved below. I wish my comments will help to improve the manuscript.

Major points

  • Line 170-171, “A single Ag particle with a 170 diameter of about 50 nm was observed at the apex of the probe”. However, it was not proved that the particle and the roughness were originated from the Ag deposition, although this is a reasonable assumption. Step-by-step SEM images or elemental mapping can prove that the particle and roughness of probe was from the Ag deposition.
  • Line 266, it is stated that “the Ag-coated TERS probe is more susceptible to damage”. Then, the power limit for the Ag-coated probe should be shown. The key point in this work is to decrease the damage on the probe. Therefore, information of power range for this probe is useful.
  • Figure 5 and Line 338, a TERS image of SWCNTs was observed, and it was explained that “the PL of Ag is a suitable criterion to assist in hotspot alignment”. However, it was not proved that the measurement was carried out at the “best” position. To demonstrate that the alignment was successfully done, the relation of PL intensitie and Raman signal should be discussed. I think, an TERS image of “misfocused condition” can strongly support the advantage of this method. For example, measurement at the half PL point could provide the weaker signal.

Minor points

  • In Figure 1, how about the HL line (orange) to camera 1? (Maybe, to simplify the system image, not necessary.)
  • Line 192, “Figure 3b”?
  • Line 343-344 can be moved to the experimental section.

Author Response

We thank the reviewer for these constructive comments of our work and appreciate the opportunity to improve the quality of our article. We have addressed all concerns raised by the reviewer as follows.

Reviewer 2 Report

Within the manuscript submitted by Xiaocong Yuan et al., photoluminescence of the silver TERS probe as a good criterion to assist in the hotspot alignment is introduced.

The paper is very interesting and deserves being published. However, before, the Authors should address the following remarks:

1. Experimental section lack of preparation method for single wall carbon nanotube sample, no information about the concentration of used solution, the solvent, etc.

2. In the introduction part large part of information concerns the enhancing properties of used properly prepared tip. However, there are no information in the Results and discussion part, concerning the enhancing properties of the presented TERS probe. As it is known, there are at least three indicators used to evaluate the performance of a TERS system, including contrast, enhancement factor (EF), and the lateral resolution of the Raman mapping. The EF is still very necessary for the research on the electromagnetic enhancement mechanism of TERS. Discussion lack of any indication about enhancement of prepared probe.

The authors try to convince:” Although the mechanism of blinking is still a subject of debate, the continuous blinking indicates that the prepared TERS probe has a stable and high electromagnetic enhancement capability because theoretical calculations have predicted that at least ~108 enhancement is necessary to observe blinking”. But, because there are no evidence for any EF calculation such statements seems to go too far and are too laconic.

3. For comparison and calibration, the conventional far-field Raman signal of the SWNTs specimen should be more informative.

4. Additionally on Fig 5 d meaning P1 and P2 (e.g. P1 stands for position) should be explained.

5. Conclusions need to be corrected.

The sentence ” We aimed to avoid unnecessary damage of precious TERS probes in our experiment.” Looks like not in right place.

And especially the lines 355-366 treat rather about SERS then TERS. There are no valid conclusions or correct conclusions about the whole work presented in the article entitled: Facilitating hotspot alignment in tip-enhanced Raman spectroscopy via the silver photoluminescence of the probe.

Author Response

We thank the reviewer for the positive evaluation of our work and appreciate the time that the reviewer spent on this manuscript. We have addressed all concerns raised by the reviewer as follows.

Reviewer 3 Report

Review on manuscript sensors-979208

“Facilitating hotspot alignment in tip-enhanced 2 Raman spectroscopy via the silver photoluminescence 3 of the probe”

by Yuan Fan, Dan Jin, XiuJu Wu, Hui Fang and Xiaocong Yuan

Recommendation

Not suitable for publication in this journal

Overall statement on the manuscript

The manuscript is written in a comprehensive form with a detailed technical description of the set-up and its performance. The introduced experimental data are fair and the conclusions are meaningful. However, a reader of a journal Sensors would expect the sensing of molecular signatures of SWCNTs in interaction with various compounds in order to understand the advantages of this new technique over standard TERS or AFM methods. So far, it is unclear, what is the benefit in the use of the TERS-AFM probe over AFM. It looks that imaging of SWCNTs by AFM is even better than by TERS-AFM set-up. In addition, this article is rather a technical report about the physical aspects of new set-up, its correction and testing than a true scientific article, in which one would demonstrate the scientific results of investigated molecular systems in the use of this set-up. Therefore, I could recommend considering this technical paper in another more appropriate journal reporting about new instruments and their technical advantages.

Best regards

Author Response

Thanks for the positive side of comments on our work. (in Action 1) Regarding the concern, now we have added some definition phrase about TERS directly related to sensing. In the reply letter, we would like to explain and stress the reason why we chose to submit the manuscript to the journal of Sensors.

Round 2

Reviewer 1 Report

As the second round of revised paper, the issues pointed are cleared. A minor errors was found (font of (d) in Figure 2), but it can be corrected in the proofreading.

I think that this manuscript is worth publication in the journal sensors.

Reviewer 3 Report

This technical article is suitable for publication.